# Evaluation of Body Composition, Physical Activity, and Food Intake in Patients with Inborn Errors of Intermediary Metabolism

**DOI:** 10.3390/nu13062111

**Published:** 2021-06-20

**Authors:** María-José de Castro, Paula Sánchez-Pintos, Nisreem Abdelaziz-Salem, Rosaura Leis, María L. Couce

**Affiliations:** 1Department of Pediatrics, University Clinical Hospital of Santiago de Compostela, 15704 Santiago de Compostela, Spain; mj.decastrol@gmail.com (M.-J.d.C.); paula.sanchez.pintos@sergas.es (P.S.-P.); 2IDIS-Health Research Institute of Santiago de Compostela, 15704 Santiago de Compostela, Spain; 3CIBERER, Instituto Salud Carlos III, 28029 Madrid, Spain; 4MetabERN, Via Pozzuolo, 330, 33100 Udine, Italy; 5Faculty of Medicine, Santiago de Compostela University, 15704 Santiago de Compostela, Spain; nisreen.abdelazizsalem@usc.es; 6CIBEROBN, Instituto Salud Carlos III, 28029 Madrid, Spain

**Keywords:** body mass index, bone mineral density, DEXA, disorders of the intermediary metabolism, height, osteopenia

## Abstract

Children with inborn errors of intermediary metabolism (IEiM) must follow special diets that restrict their intake of essential nutrients and may compromise normal growth and development. We evaluated body composition, bone mineral density, physical activity, and food intake in IEiM patients undergoing dietary treatment. IEiM patients (*n* = 99) aged 5–19 years and healthy age- and sex-matched controls (*n* = 98) were recruited and underwent dual-energy X-ray absorptiometry to evaluate anthropometric characteristics and body composition. Data on food intake and physical activity were also collected using validated questionnaires. The height z-score was significantly lower in IEiM patients than controls (−0.28 vs. 0.15; *p* = 0.008), particularly in those with carbohydrate and amino acid metabolism disorders. Significant differences in adiposity were observed between patients and controls for the waist circumference z-score (−0.08 vs. −0.58; *p* = 0.005), but not the body mass index z-score (0.56 vs. 0.42; *p* = 0.279). IEiM patients had a significantly lower total bone mineral density (BMD) than controls (0.89 vs. 1.6; *p* = 0.001) and a higher risk of osteopenia (z-score < −2, 33.3% vs. 20.4%) and osteoporosis (z-score < −2.5, 7.1% vs. 0%), but none presented fractures. There was a significant positive correlation between natural protein intake and BMD. Our results indicate that patients with IEiM undergoing dietary treatment, especially those with amino acid and carbohydrate metabolism disorders, present alterations in body composition, including a reduced height, a tendency towards overweight and obesity, and a reduced BMD.

## 1. Introduction

Intermediary inborn errors of intermediary metabolism (IEiM) are caused by genetic defects in enzymes or cofactors involved in the complex pathways by which amino acids, carbohydrates, and fatty acids are metabolized [1]. In these conditions, abnormal accumulations of substrates or deficits in the product can be detected using specific biochemical markers. Worldwide implementation of newborn screening (NBS) since the 1960s has enabled early diagnosis and treatment of IEiMs [2,3]. Treatment of IEiMs, especially aminoacidopathies (AA), organic acidemias, urea cycle disorders (UCDs), galactosemia, hereditary fructose intolerance (HFI), glycogen storage disease (GSD), and fatty acid β-oxidation defects (FAOD), mainly consists of lifelong restriction of the intake of different nutrients by limiting the amounts of natural protein, sugars, or lipids in the diet, combined with the administration of medical foods and/or supplements [4,5,6,7]. The aim of dietary treatment is to maintain metabolic stability and prevent the accumulation of toxic metabolites. Moreover, during periods of metabolic stress (e.g., intercurrent illness), acute changes in diet are regularly required to prevent metabolic decompensation, particularly in disorders that result in severe intoxication. This can result in exaggeration of the original diet, particularly in patients with AA, leading to periods of minimal protein intake and high energy intake and, ultimately, nutritional imbalance.

Regardless of dietary restrictions, the nutritional requirements necessary to ensure normal growth and development must be met in patients with IEiM [8,9]. Dietary recommendations for IEiM patients (Appendix A) are based on or extrapolated from estimated requirements for healthy populations, including recommendations from the World Health Organization (WHO) [10]. However, because IEiM diets often diverge from recommendations for natural food and energy intake, their impact on long-term growth and body composition necessitates ongoing assessment [11,12,13]. Effective treatment requires an understanding of both the biochemistry of metabolic defects and the individual nutritional requirements in order to provide an adequate intake and maintain the metabolic balance [14]. While the approach to dietary therapy is specific to each metabolic disorder, the principles are identical.

A long-standing concern is that dietary restrictions required to manage IEiM and maintain metabolic control may increase the risk of an inadequate nutrient status in both the short and long terms [14,15,16]. Dietary factors commonly linked to nutritional status include energy intake, protein quality and quantity, micronutrient intake, and the frequency and duration for which the diet must be modified during periods of increased physical activity or metabolic stress. Patients with restricted diets, especially those with low levels of natural protein, coupled with sedentary lifestyles may have the highest risk of nutritional deficits and impaired body composition and bone status [12,17,18].

In this cross-sectional observational study, we evaluated body composition, bone mineral density, physical activity, and food intake in IEiM patients and sex- and age-matched controls.

## 2. Material and Methods

### 2.1. Study Design

This cross-sectional observational study was carried out during the period 2017–2019 at the Unit of Diagnosis and Treatment of Congenital Metabolic Diseases, University Clinical Hospital of Santiago de Compostela, and included patients with IEiM who had been receiving dietary and/or medical treatment for at least 5 years. An age- and sex-matched control group of healthy children was recruited during the same period. The study was approved by the local ethics committee (registration code 2017/310), and data were collected in an encrypted manner in a database.

### 2.2. Inclusion and Exclusion Criteria

The following inclusion criteria were applied: children and teenagers (5–19 years of age) diagnosed with IEiM; written informed consent to participate in the study provided by children over 12 years old and/or parents/legal guardians of children. The exclusion criteria were as follows: patients diagnosed in the last 5 years or with unstable metabolic control; patients with another associated disease that affects physical development; and chronic treatment with anti-inflammatories, steroids, or immunosuppressant drugs for more than 3 months.

### 2.3. Study Population

A total of 99 IEiM patients (age range, 5–19 years) were recruited. The age- and sex-matched control group consisted of 98 healthy children who attended the hospital for minor surgeries. The following data were collected from all participants: diagnosis; time since diagnosis; sex; age; anthropometric measurements (weight, height, body mass index (BMI); mid-upper arm (MUAC), wrist (WrC), waist (WC), hip (HC), thigh (TC), and calf (CC) circumferences); skinfolds (biceps, triceps, subscapular, and suprailiac); puberty stage; fat mass (FM) and fat-free mass (FFM); bone densitometry of total body, lumbar spine, and proximal femur by dual-energy X-ray absorptiometry (DEXA). Dietary and pharmacological treatment and treatment compliance data were also collected, as well as the specific dietary supplements recommended in each IEiM. These supplements were included in food consumption assessment. All participants completed a physical activity questionnaire and underwent a biochemical profile analysis including metabolic biomarkers.

### 2.4. Objectives

The main study objectives were as follows: (1) assess growth and development in IEiM patients based on evaluation of anthropometric and body composition parameters; (2) determine the risk of osteopenia and/or osteoporosis based on DEXA bone mineral density measurement in different body zones; (3) evaluate food intake and physical activity using validated questionnaires.

### 2.5. Outcome Measures

#### 2.5.1. Anthropometric Assessment

Measurements were taken by the same researcher and were acquired in triplicate to minimize intra-observer bias.

-Weight was measured to the nearest 0.1 kg while the child was wearing light clothing and no shoes using a SECA 701 electronic medical scales with a class III digital display. Children were placed on the scale in a standard anatomical position without any support that could interfere with measurement. The values obtained were converted to z-scores according to the international reference values of the Centers for Disease Control and Prevention (CDC) [19].-Height was measured by using a Harpenden stadiometer (600–2100 mm), which is approved by the University of London Institute of Child Health. Height was measured while the child was standing without shoes, heavy outer garments, or hair ornaments. The values obtained were converted to z-scores according to the international reference values of the WHO [20].-BMI was calculated in kilograms per square meter (kg/m^2^), along with standardized scores and percentiles, which were scored using the international reference values of the WHO [20].-Body circumferences including MUAC, WrC, WC, HC, TC, and CC were measured with a flexible, non-extensible SECA 201 tape, which allows measurement of circumferences with millimeter precision. The tape is held at a right angle to the limb or body segment to be measured. All measures were converted to z-scores based on reference values for Spanish children and adolescents from the /enkid study [21].-Skinfold measurements (triceps, biceps, subscapular, and suprailiac skinfolds) were acquired using a Harpenden skinfold caliper. This device exerts a compression of 10 g/mm^2^ and has a measurement range of up to 80 mm in increments of 0.2. The exact point at which the skinfold measurement is taken must be carefully indicated using anatomical marking before evaluation. In this study, measurements were taken on the non-dominant side of the body [22].

#### 2.5.2. Body Composition Assessment

Body composition, including FM, muscle mass, body mass composition (BMC), and bone mineral density (BMD), was evaluated by DEXA (lunar DEXA DPX, General Electric). Measurements were taken at the lumbar level (L2, L3, and L4) and at the proximal femur. We used a z-score to compare BMD values with a population of the same sex and age, using reference values from the database published by Zanchetta et al. [20]. A BMD z-score ≤−2 is considered indicative of osteopenia risk, while a z-score ≤ −2.5 is considered indicative of osteoporosis risk. FM and FFM as determined by DEXA were converted to z-scores using reference values for the Spanish population [22].

#### 2.5.3. Physical Activity and Feeding Questionnaires

The International Physical Activity Questionnaire (IPAQ) [22] was used to collect information on physical activity habits in general, and on different physical activity patterns, both programmed and spontaneous. Data were also collected on moderate and vigorous/intense activity (<7 h or ≥7 h per week, and number of days of vigorous exercise per week). All participants completed a 3-day food consumption survey in which the total amount of every meal component was measured in grams. These data were then analyzed using computer software to calculate the average dietary intake over the 3-day period.

#### 2.5.4. Biochemical Analysis

Blood samples were collected by venous puncture after fasting for at least 12 h, except in FAOD patients and some carbohydrate metabolism disorder (CHD) patients, from whom samples were acquired after fasting for 6 h. No intense physical activity was allowed in the hour before blood extraction. The following serum parameters were determined (reference values are shown in parentheses): total cholesterol (120.0–255.0 mg/dL); high-density lipoprotein cholesterol (HDL-c; 34.0–91.0 mg/dL); low-density lipoprotein cholesterol (LDL-c; 55.0–125.0 mg/dL); triglycerides (27.0–150.0 mg/dL); total proteins (6.4–8.5 g/dL); albumin (4.4–5.6 g/dL); urea (14.0–43.0 mg/dL); creatinine (0.4–1.1 mg/dL); calcium (9.0–10.5 mg/dL); folate (2.7–17 ng/mL); sodium (134.0–145.0 mmol/L); glucose (74.0–105.0 mg/dL); vitamin D 25(OH)D and 1–25(OH)2D (deficiency, ≤10 ng/mL; insufficiency, 10–20 ng/mL; recommended, >20 ng/mL); vitamin A (1–6 years, 0.20–0.43 mg/L; 7–12 years, 0.26–0.49 mg/L; 13–19 years, 0.26–0.72 mg/L; >19 years, 0.30–0.80 mg/L); vitamin E (1–12 years, 0.30–0.90 mg/dl; 13–19 years, 0.60–1.00 mg/dL; >19 years, 0.50–1.81 mg/dL); vitamin K1 (0.10–2.10 ng/mL); zinc (65–140 μg/dL); and selenium (60–150 μg/L).

### 2.6. Statistical Analyses

Data analyses were performed using the statistical package SPSS Statistics 22 (IBM). A descriptive statistical study of the sample was performed, analyzing the parameters of centralization (mean and median) and dispersion (standard deviation, maximum and minimum, quartiles, and range) and position parameter percentiles 5, 10, 25, 50, 75, 85, 90, 95, and 99. Dummy variables were used for categorical variables that had more than two categories. Univariate (frequencies and proportions) and bivariate (contingency tables) statistical methods were used. The chi-square test was used to detect statistically significant differences. Pearson’s and Spearman’s bivariate correlations were also calculated. The mean and standard deviation (SD) were used as measures of central tendency and variance for all primary outcome variables: height z-score, weight z-score, BMI z-score, % FM, calories, protein (g/kg and % of calories), fat (% of calories), and carbohydrate (% of calories). A paired t-test was used to compare body composition variables between the patient and control groups. We used 95% confidence intervals for the mean outcomes of the primary variables.

## 3. Results

### 3.1. General Characteristics of the Study Population

The study population consisted of 197 individuals (99 IEiM patients and 98 age- and sex-matched controls). Females accounted for 57.6% and 51.1% of the patient and control groups, respectively. There were no significant differences in sex distribution between groups. Within the patient group, 94% received an early diagnosis following newborn screening (NBS), while the remaining 6% received a delayed diagnosis. AAs were the most prevalent of all IEIMs, accounting for 77.8% (*n* = 77) of cases, followed by CHDs (12.1%, *n* = 13) and FAODs (10.1%, *n* = 10). The AA with the highest incidence was phenylketonuria (PKU; 26 females and 20 males), followed by hypermethioninemia due to MAT I-III deficiency (4 females and 5 males), maple syrup urine disease (MSUD; 2 females and 2 males), and methylmalonic acidemia (MMA; 2 females and 2 males) (Table 1).

### 3.2. Assessment of Anthropometric Characteristics

#### 3.2.1. Anthropometric Characteristics: IEiM Patients vs. Controls

BMI, height, and WC percentiles and z-scores for IEiM patients and controls are shown in Table 2.

Analysis of BMI revealed significant differences between patients and controls for the following parameters: overweight/obesity (35.4% of patients vs. 30.6% of controls); BMI in the *p* > 95 range (25.2% of patients vs. 6.1% of controls); underweight (8.1% of patients vs. 4.1% of controls). After adjusting for sex, the frequency of underweight was higher in female than male IEiM patients (8.8% vs. 7.1%; *p* < 0.05). No significant differences in the frequency of overweight were observed between male and female IEiM patients.

Mean height percentiles and z-scores were significantly lower in IEiM patients than controls. Significant differences were observed between patients and controls for height in the *p* < 5 range (11 patients vs. 0 controls). No significant differences were observed for height in the *p* > 95 range (seven patients vs. five controls). Significant differences were observed between groups for WC percentiles and z-scores, which were higher in patients than controls (WC in the *p* > 95 range, 10 patients vs. 0 controls; WC in the *p* 85–95 range, 11 patients vs. 5 controls). Comparable findings were observed for the corresponding z-scores.

For all other anthropometric measurements, no significant differences were observed between groups (Appendix A).

#### 3.2.2. Anthropometric Characteristics: Intermediary Metabolism Disorders

After stratification of patients according to individual IEiM, overweight was more prevalent in patients with AA (36.4%), and underweight in CHD (16.7%) (Table 3).

Significant differences were also observed for height and BMI z-scores. Patients with CHD had the lowest mean height z-score, followed by patients with AA. Significant differences were observed for the BMI z-score, which was higher in AA and FAOD. No significant differences were observed for body circumference parameters, except for WC, which was higher in AA patients (*p* = 0.0001) (Table 4).

### 3.3. Body Composition Assessment

#### 3.3.1. Body Composition Assessment: IEiM Patients vs. Controls

DEXA revealed no significant differences in MM, FM, or BMD in the lumbar spine (L2–L4) between IEiM patients and controls. However, mean total body and proximal femur BMDs were significantly lower in patients than controls. These differences persisted after adjusting for sex (Table 5).

In IEiM patients, Pearson’s correlation coefficient (r) revealed a significant (*p* < 0.01) positive correlation between bone variables (BMD of the spine, femoral trochanter, femoral ward, and femur) and both BMI and weight.

There were significant differences in osteopenia risk between IEiM patients and controls (*p* = 0.036), although no history of fracture was recorded in either group. Risk of osteopenia was observed in 33 patients (33.3%) vs. 20 (20.4%) controls, and a very low BMD (z-score < −2.5) was found in 7 (7.1%) patients vs. 0 controls.

#### 3.3.2. Body Composition Assessment: IEiM Subtypes

Stratification of body composition parameters according to IEiM subgroup revealed a significantly higher FM z-score and significantly lower total body and femur BMD z-scores in AA patients than in those with an HCD or FAOD (*p* < 0.05) (Table 6). The risks of osteopenia and osteoporosis were highest in the CHD (50% and 1.7%, respectively) and AA (32.5% and 6.5%, respectively) subgroups.

### 3.4. Assessment of Patterns of Physical Activity

The proportion of participants that did not engage in vigorous physical activity 3 days a week was 89% in the patient group vs. 65% in the control group (*p* = 0.0001). Compliance with WHO recommendations for moderate intense physical activity was observed in 10% of patients vs. 19% of controls (*p* = 0.041).

In IEiM patients, we observed a positive correlation between moderate and vigorous physical exercise (measured in minutes/week) and muscle mass (r = 0.314, *p* = 0.002; r = 0.212, *p* = 0.035). Pearson’s coefficient revealed a significant positive correlation between both forms of physical activity and femur BMD in IEiM patients.

### 3.5. Food intake Assessment

#### 3.5.1. Food Intake: IEiM Patients vs. Controls

Protein intake was significantly lower in IEiM patients than controls (55.75 ± 21.23 vs. 75.67 ± 4.61 g/day; *p* = 0.000). Similarly, protein energy, protein percentage, and total protein per kg body weight were significantly lower in IEiM patients than controls (*p* < 0.05) (Figure 1).

IEiM patients consumed significantly more carbohydrates than controls (total intake, 234.57 ± 119.76 vs. 201.79 ± 39.26 g/day; *p* = 0.013), with a percentage of carbohydrate energy from the total calorie intake ranging between 46 and 78%. There were no significant differences between groups in dietary intake of fat or energy (Appendix A). Comparison of mineral intake between IEiM patients and controls revealed comparable values for potassium, calcium, magnesium, phosphorus, iron, selenium, and zinc. Folate intake was significantly higher in patients than controls (283.3 ± 155.3 vs. 226.13 ± 165.3 µg/L; *p* = 0.015), and there were no significant differences between groups in the intake of fat-soluble vitamins D, K, and E. Mean vitamin A intake was higher in IEiM patients than controls (527.28 vs. 440.7 mg/L; *p* = 0.047) (Appendix A).

We observed a significant positive correlation between total body BMD and both total protein and natural protein intake (r = 0.186, r = 0.254; *p* < 0.01). Pearson’s coefficient revealed a significant positive correlation between protein intake and BMD measured at the trochanter (r = −0.177, *p* = 0.014) and Ward’s triangle (r = 0.171, *p* = 0.018). There was also a significant positive correlation between natural protein intake and BMD measured at the femur (r = 0.213, *p* = 0.003), trochanter (r = 0.257, *p* = 0.000), and Ward’s triangle (r = 0.233, *p* = 0.001). No correlation was observed between BMD and either calcium or vitamin D intake.

#### 3.5.2. Food Intake: IEiM Subtypes

Daily dietary intake varied across IEiM subtypes. The lowest protein intake was observed in patients with AA (50.38 g/day vs. 75.67 g/day in controls; *p* = 0.000), and the highest fat intake was observed in those with CHD (67.21 g/day vs. 79.17 g/day in controls; *p* = 0.821). CH intake was significantly higher in AA and CHD patients compared with controls. No significant differences in total energy intake per day were observed between any IEiM subtype and controls (Table 7).

## 4. Discussion

### 4.1. Anthropometric Characteristics

Key findings of the present study include the significantly higher BMI z-scores in AA and FAOD patients compared with controls, and the significantly lower height z-scores in IEiM patients, particularly those with AA or CHD, compared with controls. By contrast, no significant difference in the BMI z-score was found in CHD patients compared with controls. Several authors have reported significantly lower mean height z-scores in children and adults with classic GSD [23,24] and galactosemia [25,26,27], but there is less information regarding this condition in HFI [28]. Trace amounts of dietary fructose and quantitative or qualitative dietary deficits linked to dietary restrictions are among the proposed causes of height deficits in HFI patients [28]. In line with this hypothesis, the subgroup of HFI patients in our study population followed a fructose-, sucrose-, and sorbitol-restricted diet, but six had a daily fructose intake that exceeded the recommended 2 g/day. In GLUT1 deficiency, a recent revision showed that 10% of the patients, all under 10 years old, had a length/height z-score below −1.6 [29]. The possible influence of a ketogenic diet on the anthropometric evolution in GLUT1 deficiency patients, although it is the subject of debate, does not seem to be a relevant factor in most patients [30]. In AA patients, we observed significantly lower height, BMI, WC, and biceps skinfold z-scores compared with controls. There findings build upon existing knowledge of alterations in body composition parameters in children with PKU, MMA, propionic academia (PA), and UCD [12,31,32,33,34,35,36,37,38,39,40]. Optimization of certain plasma amino acid levels (L-arginine and L-valine levels in MMA and PA; and L-leucine and L-valine in UCD) and an adequate protein-to-energy intake ratio are essential to support normal growth in AA disorders [18].

FAOD patients had the highest weight and height z-scores of all IEiM subtypes and markedly higher BMI values compared with controls (*p* = 0.014). This may be explained by the low-fat, high-carbohydrate diet followed by FAOD patients, who also require frequent meals to prevent hypoglycemia. Despite the recognized role of impaired free fatty acid mitochondrial oxidation in the pathogenesis of obesity and insulin resistance [41,42], little is known about the progression of BMI in FAOD.

### 4.2. Body Composition Assessment

We observed significantly higher FM z-scores in IEiM patients than controls (12.25 ± 8.11 vs. 10.03 ± 6.45 kg; *p* = 0.040). Subgroup analysis showed that this was mainly due to a higher mean FM z-score among AA patients vs. controls (0.315 ± 1.83 vs. 0.071 ± 1.35 kg; *p* < 0.05). No significant differences in FM were observed between CHD or FAOD patients and controls.

The association between body composition and bone health is a topic of debate. In children, adipose tissue may stimulate bone growth [43], and a positive association between FM and bone mass has been reported [44,45]. Nonetheless, in our series, despite significant positive correlations between weight and BMI measured at almost all bone sites, total body BMD and BMD measured at the femoral neck, femoral trochanter, and Ward’s triangle were lower in IEiM patients than healthy controls. Factors proposed to contribute to reduced bone mineralization in IEiM include nutritional deficiencies linked to the dietary regimens of patients with these diseases, reduced physical activity and sunlight exposure, and early ovarian failure [11]. We observed a significant negative correlation between age and BMD measured in the lumbar spine, femur, and trochanter.

The total body BMD z-score was lower in AA patients than controls (0.833 ± 0.92 vs. 1.66 ± 1.35; *p* = 0.000), in agreement with the findings of BMD in previous studies conducted in patients with PKU, UCD, and MMA [36,46,47,48,49]. Data of factors that influence BMD for this group of pathologies, with the exception of PKU [14,50], are limited. The impaired amino acid homeostasis in AA could influence the balance of bone remodeling, favoring a trend towards increased bone resorption in line with PKU patients [14].

In our study cohort, BMD values were lower in CHD patients than controls, although this effect did not reach significance. This is in line with previous reports of low BMD values in children and adults with classical galactosemia [51,52] and in GSD Ia and Ib, GSD II, GSD III, GSD V, and GSD IX patients [53,54,55]. The low BMD in CHD patients could be due to their particular diet, which can result in deficiencies in certain nutrients necessary for a healthy bone density (e.g., calcium and vitamin D), due to the metabolic disorder itself [56,57], or as a consequence of poor metabolic control, as it has been suggested in GSD type 1 [58,59].

### 4.3. Physical Activity in IEiM Patients

Another important finding in our study population was a significantly lower level of physical activity in IEiM patients than controls. Most IEiM patients performed less vigorous exercise (<3 days/week and <7 h/week) than controls and failed to comply with WHO recommendations for physical activity [60]. Physical activity is of undisputed importance for personal and social development [61] and is a key determinant of body composition, especially FM, and BMD. We found that moderate-to-vigorous physical activity was positively correlated with spine and femur bone density and muscle mass. To date, no studies have evaluated the effect of physical activity on body composition and BMD in IEiM patients.

The risks posed by physical activity in IEiM patients vary according to the specific metabolic disorder. For example, prolonged or intense physical exercise is a recognized potential trigger of hyperammonemic crises in patients with urea cycle disorders [6]. By contrast, the practice of sport implies no additional risk in PKU patients provided caloric intake and protein catabolism are monitored [62], even though there is no consensus on recommended protein supplementation for PKU patients practicing sports [63]. Similarly, sports pose no additional risk to patients with classic galactosemia, even though these patients may be less physically active due to motor dysfunction [64,65]. Exercise intolerance is a characteristic feature in muscular GSD [66], caused by an increase in glycogen storage that disrupts contractile function and/or reduced substrate turnover, which inhibits skeletal muscle ATP production [67]. Exercise intolerance is frequent in patients with FAO disorders, as energy homeostasis during fasting or prolonged exercise depends on mitochondrial fatty acid oxidation. Moreover, in individuals with FAO defects physical exercise may be contraindicated or limited due to possible cardiovascular complications caused by the disease.

### 4.4. Dietary Intake in IEiM Patients

The diet and type of energy consumed differed between patients and controls. The daily intake of total protein, natural protein, and energy from protein was significantly lower while that of carbohydrates was significantly higher in IEiM patients than controls. A cardinal principle in the dietary management of children of AA and FAOD is to prevent lapse into a catabolic state by maintaining an aggressive nutrition regimen that promotes a positive nitrogen balance and decreases the risk of metabolic decompensation and accumulation of toxic metabolites [68]. As expected, given the dietary recommendations in AA, the intake of total protein and natural protein was lowest in AA patients. Our data support a significant positive relationship between protein intake and muscle mass, and between total protein intake and total body BMD. However, bone health depends not only on quantitative protein intake [69] but also on protein structure quality [70]. We found that natural protein intake was positively and significantly correlated with BMD measured in the femur (r = 0.208, *p* = 0.003), trochanter (r = 0.264, *p* = 0.000), and Ward’s triangle (r = 0.287, *p* = 0.001). Another notable finding is the significantly higher intake of CH vs. fat in AA patients (237.63 ± 127.95 and 47.67 ± 18.11 g, respectively), with the potential associated risk of carbohydrate intolerance and insulin resistance, as documented in PKU patients [41].

In our study population, we observed no correlation between bone mineral density and either calcium or vitamin D intake, and the intake of minerals such as calcium, potassium, phosphorous, iron, and zinc was normal.

A limitation of our study is the absence of patients with long-chain beta-oxidation disorders, inclusion of whom would have complemented the specific analysis of physical activity. Strengths of this study include its comprehensive and novel analysis of body composition in a large and representative sample of IEiM, providing relevant aspects for the management of these patients.

## 5. Conclusions

Patients with IEiM are at risk of impairment of physical development and bone health status caused by the restrictive dietary regimes required to manage these conditions. Compared with matched healthy controls, IEiM patients in our study had a significantly lower height (particularly those with CHD), a higher FM (kg), and a markedly lower BMD. In AA patients, we observed significant decreases in height, BMI, WC, and biceps z-scores. Interestingly, the level of physical activity was lower in IEiM patients. Moderate-to-vigorous physical activity was positively correlated with bone mineral density and muscle mass, suggesting that regular physical activity may play a key role in optimizing body composition in IEiM patients.

## Figures and Tables

**Figure 1 nutrients-13-02111-f001:**
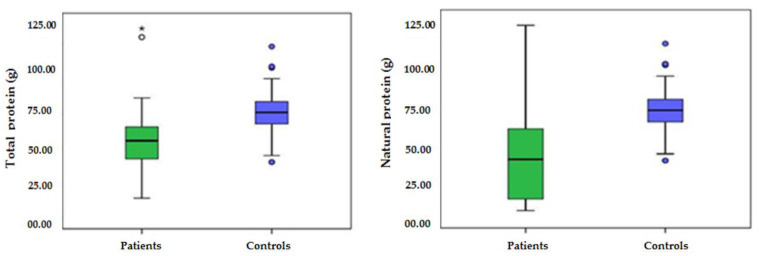
Protein intake in IEiM patients vs. controls.

**Table 1 nutrients-13-02111-t001:** IEIM subtypes in participating patients, according to sex.

	Patients			
	Females		Males	
	N	%	N	%
Aminoacidopathies				
Mild HPA	12	27.9	9	26.5
PKU	14	32.5	11	32.4
Hypermethioninemia by MAT I/III deficiency	4	9.3	5	14.7
MSUD	4	9.3	2	5.8
Tyrosinemia type 1	2	4.7	2	5.8
Glutaric aciduria type 1	3	7	3	8.8
Citrulinemia type I	1	2.3	0	0
OTC deficiency	0	0	1	2.9
3-Hydroxy-3-methylglutaric aciduria	0	0	1	2.9
Methylmalonic aciduria	2	4.7	0	0
Nonketotic hyperglycinemia	1	2.3	0	0
Carbohydrate disorders and defects of transport of carbohydrates				
Classic galactosemia	2	28.6	1	16.7
Hereditary fructose intolerance	2	28.6	3	50
Glycogen storage disease type 1	2	28.6	2	33.3
Glucose transporter 1 deficiency	1	14.2	0	0
Fatty acid β-oxidation disorders				
MCADD	3	42.9	3	100
SCADD	4	57.1	0	0

HPA, hyperphenylalaninemia; MCADD, medium-chain acyl-CoA dehydrogenase deficiency; OTC, ornithine transcarbamylase; PKU, phenylketonuria; MSUD, maple syrup urine disease; SCADD, short-chain acyl-CoA dehydrogenase deficiency.

**Table 2 nutrients-13-02111-t002:** Distribution of body mass index, height, and waist circumference percentiles and z-scores, based on WHO international standards.

	Patients		Controls		
	N	%	N	%	*p*
**BMI PERCENTILE**					
Underweight (*p* < 5)	8	8.1	4	4.1	**0.039**
Normal weight (P5–85)	56	56.6	64	65.3	**0.000**
Overweight and obesity (*p* > 85)	35	35.4	30	30.6	**0.000**
**BMI Z-SCORE**					
−3.090 to ≤ −1.645	8	8.08	4	4.1	**0.044**
−1.645 to ≤ 1.030	56	56.6	64	65.3	**0.037**
1.030 to ≤ 1.645	9	9.09	25	25.5	**0.017**
1.645 to 3.090	24	24.2	5	5.1	**0.044**
≥3.090	2	2.02	0	0	0.614
**HEIGHT PERCENTILE**					
*p* < 5	11	11.1	0	0	**0.001**
P5–95	81	81.8	93	94.9	**0.007**
*p* > 95	7	7.1	5	5.1	0.812
**HEIGHT Z-SCORE**					
≤2	8	8.1	0	0	**0.001**
−2 to −1	15	15.2	12	12.2	0.511
−1 to 1	61	61.6	64	65.4	0.322
1 to 2	10	10.1	17	17.3	0.056
≥2	5	5	5	5.1	0.849
**WC PERCENTILE**					
P10–85	62	62.6	64	65.3	0.884
P85–95	11	11.1	5	5.1	**0.035**
*p* > 95	10	10.1	0	0	**0.002**
**WC Z-SCORE**					
≤−2	6	6.1	11	11.2	**0.041**
−2 to −1	23	24.2	23	23.5	0.741
−1 to 1	49	48.5	58	60.2	**0.037**
1 to 2	12	12.1	6	5.1	**0.047**
≥2	9	9.1	0	0	**0.003**
Total	99	100	98	100	

BMI, body mass index; WC, waist circumference. Differences considered significant at *p* < 0.05. In bold when data are significant

**Table 3 nutrients-13-02111-t003:** Degree of adiposity in intermediary metabolism disorders according to WHO international standards.

BMI Category	AA		FAOD		CHD	
	N	%	N	%	N	%
Underweight (*p* < 5)	5	6.5	1	10	2	16.7
Normal weight (P5–85)	44	57.1	6	60	6	50
Overweight and obesity (*p* > 85)	28	36.4	3	30	4	33.3
Total	77	100	10	100	12	100

AA, aminoacidopathies; BMI, body mass index; CHD, carbohydrate disorders; FAOD, fatty acid β-oxidation defects.

**Table 4 nutrients-13-02111-t004:** Mean body composition z-scores among intermediary metabolism disorders compared with controls.

		Patients		Controls		
Anthropometric Parameter (z-Score)	IEiM Subtype	N	Mean ± SD	N	Mean ± SD	*p*
Weight	AA	77	0.469 ± 1.12	98	0.486 ± 0.86	0.913
	FAOD	10	0.935 ± 0.98	20	0.297 ± 0.94	0.136
	CHDs	12	−0.349 ± 1.39	20	0.248 ± 0.76	0.179
Height	AA	77	−0.267 ± 1.18	98	0.148 ± 0.95	**0.012**
	FAOD	10	0.711 ± 1.16	20	−0.13 ± 0.82	0.061
	CHD	12	−1.173 ± 1.04	20	−0.03 ± 0.92	**0.007**
BMI	AA	77	0.662 ± 1.22	98	−0.373 ± 0.95	**0.000**
	FAOD	10	0.64 ± 1.33	20	−0.68 ± 0.98	**0.014**
	CHD	12	−0.033 ± 1.73	20	−0.379 ± 0.83	0.447
Mid-arm circumference	AA	77	−0.242 ± 1.14	98	−0.373 ± 0.99	0.417
	FAOD	10	−0.15 ± 1.3	20	−0.68 ± 0.98	0.287
	CHD	12	−0.9 ± 1.45	20	−0.379 ± 0.83	0.339
Waist circumference	AA	77	−0.075 ± 1.35	98	−0.588 ± 1.11	**0.008**
	FAOD	10	−0.285 ± 1.57	20	−0.794 ± 1.09	0.382
	CHD	12	0.052 ± 1.2	20	−0.363 ± 0.69	0.251
HIP circumference	AA	77	−0.635 ± 1.08	98	−0.821 ± 0.88	0.319
	FAOD	10	0.219 ± 1.19	20	−1.014 ± 0.83	**0.080**
	CHD	12	−1.16 ± 1.23	20	−0.778 ± 0.84	0.443
Biceps skinfold	AA	77	0.778 ± 1.57	98	0.288 ± 1.44	**0.036**
	FAOD	10	0.241 ± 2.44	20	0.109 ± 1.28	0.672
	CHD	12	0.443 ± 1.52	20	0.787 ± 125	0.726
Triceps skinfold	AA	77	0.131 ± 1.25	92	0.015 ± 1.4	0.547
	FAOD	10	−0.215 ± 1.59	12	0.022 ± 1.33	0.708
	CHD	12	−0.052 ± 1.28	15	0.298 ± 1.03	0.514
Subscapular skinfold	AA	77	1.1 ± 2.38	98	0.578 ± 1.79	0.106
	FAO	10	1.293 ± 1.95	20	0.82 ± 1.84	0.811
	CHD	12	0.817 ± 2.15	20	0.792 ± 1.46	0.930
Suprailiac skinfold	AA	77	1.38 ± 1.86	98	1.03 ± 1.66	0.201
	FAOD	10	1.293 ± 1.95	20	0.82 ± 1.84	0.565
	CHD	12	1.363 ± 2.04	20	1.176 ± 1.12	0.755

AA, aminoacidopathies; BMI, body mass index; CHD, carbohydrate disorders; FAOD, fatty acid oxidation disorders; IEiM, inborn errors of intermediary metabolism; N, number of participants; SD, standard deviation. Differences considered significant at *p* < 0.05. In bold when data are significant

**Table 5 nutrients-13-02111-t005:** Mean body composition z-scores as measured by DEXA in patients and controls.

	Patients	Controls	
z-Score	Mean ± SD	Mean ± SD	*p*
Muscle mass	0.09 ± 1.6	0.29 ± 1.3	0.317
Fat mass	0.26 ± 1.8	−0.07 ± 1.3	0.159
BMD total	0.89 ± 0.95	1.6 ± 1.5	**0.001**
Lumbar spine L2-L4	0.65 ± 0.69	0.8 ± 0.64	0.117
Femur neck	0.45 ± 0.76	0.67 ± 0.75	**0.044**
Femur trochanter	0.5 ± 0.98	0.84 ± 0.89	**0.012**
Ward’s triangle	−0.21 ± 0.71	0.04 ± 0.8	**0.023**

BMD, bone mineral density; SD, standard deviation. Differences considered significant at *p* < 0.05. In bold when data are significant

**Table 6 nutrients-13-02111-t006:** Mean body composition z-scores measured by DEXA according to IEiM subtype.

		Patients		Controls		
z-Score	IEiM Subtype	N	Mean ± SD	N	Mean ± SD	*p*
Muscle mass	AA	77	0.023 ± 1.39	98	0.297 ± 1.26	0.383
	FAOD	10	1.515 ± 2.3	20	0.24 ± 1.16	0.110
	CHD	12	−0.687 ± 1.39	20	0.046 ± 1.24	0.254
Fat mass	AA	77	0.315 ± 1.83	98	−0.071 ± 1.35	**0.026**
	FAOD	10	−0.021 ± 2.34	20	−0.35 ± 1.22	0.676
	CHD	12	0.162 ± 1.65	20	0.086 ± 0.73	0.940
BMD total	AA	77	0.833 ± 0.92	98	1.66 ± 1.56	**0.000**
	FAOD	10	1.38 ± 0.76	20	1.493 ± 1.06	0.780
	CHD	12	0.863 ± 1.26	20	1.639 ± 1.63	0.213
BMD spine L2–L4	AA	77	0.641 ± 0.68	98	0.88 ± 0.64	0.321
	FAOD	10	1.024 ± 0.59	20	0.935 ± 0.78	0.818
	CHD	12	0.38 ± 0.71	20	0.88 ± 0.76	0.206
BMD femur trochanter	AA	77	0.491 ± 095	98	0.84 ± 0.89	**0.013**
	FAOD	10	1.06 ± 1.16	20	0.45 ± 1.25	0.753
	CHD	12	0.11 ± 0.85	20	0.49 ± 1.15	0.350
BMD femur neck	AA	77	0.424 ± 0.79	98	0.67 ± 0.75	**0.039**
	FAOD	10	0.528 ± 1.33	20	0.867 ± 0.76	0.473
	CHD	12	0.26 ± 0.56	20	0.36 ± 0.89	0.733
BMD Ward’s triangle	AA	77	−0.252 ± 0.71	98	0.04 ± 0.8	**0.013**
	FAOD	10	0.283 ± 0.66	20	−0.25 ± 1.2	0.229
	CHD	12	−0.33 ± 0.59	20	−0.33 ± 0.84	0.994

AA, aminoacidopathies; BMD, bone mineral density; CHD, carbohydrate disorders; FAOD, fatty acid oxidation disorders; IEiM, inborn errors of intermediary metabolism; N, number of participants; SD, standard deviation. Differences considered significant at *p* < 0.05. In bold when data are significant.

**Table 7 nutrients-13-02111-t007:** Mean dietary intake/day according to IEiM subtype.

	Patients			Controls		
Dietary Intake/Day	IEiM Subtype	N	Mean ± SD	N	Mean ± SD	*p*
Protein, total (g)	AA	77	50.38 ± 15.9	98	75.67 ± 14.6	**0.000**
	FAOD	10	83.29 ± 27.26	20	74.55 ± 14.97	0.753
	CHD	12	67.21 ± 24.9	20	79.17 ± 15.3	0.821
Protein, natural (g)	AA	77	32.18 ± 23.57	98	75.67 ± 14.6	**0.000**
	FAOD	10	83.29 ± 27.26	20	74.55 ± 14.97	0.658
	CHD	12	67.21 ± 24.9	20	79.17 ± 15.3	0.835
Protein (g/kg)	AA	77	1.23 ± 0.45	98	2.08 ± 0.89	**0.000**
	FAOD	10	2.03 ± 0.58	20	2.297 ± 0.95	0.432
	CHD	12	2.08 ± 1.02	20	1.99 ± 0.64	0.421
Fat, total (g)	AA	77	47.67 ± 18.11	98	47.73 ± 10.86	0.954
	FAOD	10	38.09 ± 13.11	20	49.04 ± 9.27	**0.033**
	CHD	12	60.48 ± 32.24	20	44.58 ± 9.3	**0.002**
CH, total (g)	AA	77	237.63 ± 127.95	98	201.79 ± 39.26	**0.012**
	FAOD	10	243.73 ± 94.66	20	209.92 ± 41.22	0.654
	CHD	12	216.12 ± 78.46	20	199.31 ± 34.55	**0.004**
Energy, total (Kcal)	AA	77	1541.75 ± 386.65	98	1540.83 ± 224.77	0.945
	FAOD	10	1695.64 ± 480.22	20	1587.58 ± 237.04	0.854
	CHD	12	1678.12 ± 406.34	20	1516.69 ± 200.7	**0.027**
Energy (Kcal/kg)	AA	77	39.34 ± 16.39	98	43.09 ± 19.49	0.843
	FAOD	10	42.4712.89	20	49.12 ± 17.88	0.895
	CHD	12	52.48 ± 22.56	20	38.67 ± 13.37	0.756

AA, aminoacidopathies; CHD, carbohydrate disorders; FAOD, fatty acid β-oxidation disorders; IEiM, inborn errors of intermediary metabolism; N, number of cases. Differences considered significant at *p* < 0.05. In bold when data are significant

## Data Availability

Data are found in archive data of Metabolic Unit of Universitaruy clinical hospital of Santiago de Compostela.

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
