# Peer review of "Evaluation of Body Composition, Physical Activity, and Food Intake in Patients with Inborn Errors of Intermediary Metabolism"

_nutrients, 2021, doi:10.3390/nu13062111_

Round 1

Reviewer 1 Report

The work by de Castro and colleagues describes the impact of inborn errors of intermediary metabolism (IEiM) on anthropometrics of two human age (5-19 years old) and sex-matched cohorts of patients harboring different kind of IEiMs, namely aminoacidopathies (AA) and carbohydrate disorders, versus healthy controls.
This largely descriptive study is well designed, written and the results are useful and illustrative of the impact that dietary restrictions provoked by IEiMs have on height, obesity and bone mineral density. It also illuminates which physiological metrics are affected by specific metabolic areas affected by the IEiMs.
This Reviewer has any concern about this work and can be published as is.

Author Response

Thank you very much for your comments.

       Reviewer 2 Report

I was pleased to read the proposed article by De Castro et al. entitled "
Evaluation of Body Composition, Physical Activity, and Food Intake in Patients with Inborn Errors of Intermediary Metabolism". The article is interesting and well written. I only have a few suggestions on how the article can be improved.

1. To facilitate understanding of the results, particularly on how dietary restrictions may impact the outcomes of interest, I suggest including a table with the overall dietary recommendations specific to each disorder included in the study.

2. Were nutrient supplements included in the assessment of food consumption? Please specify.

In the discussion section:
3. Lines 318-320: A subject with GLUT1-deficiency is also present in the sample. Also in this population, the issue of lower height is a debated topic. Please add. 

4. Anthropometric characteristics section: Give a mechanistic explanation of how different dietary restrictions impact reduced height/increased BMI.

5. Body composition section: Give a mechanistic explanation of how dietary restrictions in AA impact body composition.

Minor comments:
- The order of the authors in the submission is different from that in the text. Please check
- Table 2, BMI Z-score, second line, correct 1.030 with 1.036

Author Response

  1. To facilitate urderstanding of the results, particularly on how dietary restrictions impact the outcomes of interest, I Suggest including a table with the overall dietary recommendations specific to each disorder included in the study,

ANSWER: We thank the suggestion and we add a supplementary table 1.

  1. Were nutrients supplements included in the assessment of food consumption?

ANSWER: Yes, they were included. To clarify this point we modify the manuscript as detailed below:

Line 100: “Dietary and pharmacological treatment and treatment compliance data were also collected, as well as, the specific dietary supplements recommended in each one of the IEiM. These supplements were included in food consumption assessment. All participants completed a physical activity questionnaire and underwent a biochemical profile analysis including metabolic biomarkers”.

3- Lines 318-320_ A subject with GLUT-1 deficiency is also present in the sample. Also in this population, the issue of lower height is a debated topic. Please add.

ANSWER: In Anthropometric characteristics section in the discussion we include the next paragraph and two new references (number 29 and 30, detailed below):

                Line 307: In GLUT1 deficiency a recent revision showed that 10% of the patients, all under 10 years old, had length/height below -1.6 z-score (29). The possible influence of ketogenic diet in anthropometric evolution in GLUT1 deficiency patients, although it is the subject of debate, does not seem to be a relevant factor in most patients (30).

  1. Bertoli S, Masnada S, De Amicis R, Sangiorgio A, Leone A, Gambino M, et al. Glucose transporter 1 deficiency syndrome: nutritional and growth pattern phenotypes at diagnosis. Eur J Clin Nutr. 2020;74(9):1290-1298.
  2. Ferraris C, Guglielmetti M, Pasca L, De Giorgis V, Ferraro OE, Brambilla I,et a. Impact of the Ketogenic Diet on Linear Growth in Children: A Single-Center Retrospective Analysis of 34 Cases. Nutrients. 2019;11(7):1442.2.
  3. Anthropometric characteristics section: Give a mechanistic explanation of how different dietary restrictions impact reduced height/increased BMI

ANSWER: Obviously, as the different subgropus of IEiM included, have completely different pathogenesis and enzimatic, the pathogenic factors are not uniform.

                In AA the protein-to-energy intake ratio and the plasma level of some amino acids in certain disorders (L-arginine and L-valine levels in methylmalonic and propionic acidemia; and L-leucine and L-valine in UCD) were positively associated with height (Molema F et al., 2019). The usual practice in our unit, according to recommentations, comprises regular monitoring of plasma amino acids, in order to adapt in an evolutionary way the natural protein intake and the supplement of essential amino acids to protein tolerance, clinical stability and plasma levels of amino acids.

                In FAOD patients, as already explained in this section, the higher weight and height z-scores of all IEiM subtypes and markedly higher BMI values compared with controls may be explained by the low-fat and high-carbohydrate diet followed by FAOD patients, with the highest energy intake of the three sybgroups of IEiM and higher (not signifficant) the energy intake in controls.

                Between CHD patients, in addition to the referred peculiarities of patients with HFI and galactosemia, in some type of GSD, growth failure could be linked in part to hepatic resistance to GH (Brooks ED et al, 2013).

We include this paragraph in this section (line 313): “Optimization of certain plasma amino acid levels (L-arginine and L-valine levels in MMA and PA; and L-leucine and L-valine in UCD), and an adequate protein-to-energy intake ratio are essential to support normal growth in AA disorders (18)”.

  1. 5. Body composition section: Give a mechanistic explanation of how dietary restrictions in AA impact body composition.

ANSWER: an optimal balance among amino acids in the diet is crucial for whole body homeostasis. There is growing recognition that some amino acids regulate key metabolic pathways that are necessary for maintenance, growth, reproduction, and immunity. They are called functional amino acids, which include arginine, cysteine, glutamine, leucine, proline, and tryptophan.

In AA the amino acid homeostasis is impaired, and this aspect could influence the balance of bone remodeling favoring a trend towards increased bone resorption in line with findings in PKU patients.

We add the next explanation in line 340: “The impaired amino acid homeostasis in AA could influence the balance of bone remodeling favoring a trend towards increased bone resorption in line with PKU patients (14)”.

  1. The order of the authors in the submission is different from that in the text. Please check

ANSWER: The correct order is the one reflected in the manuscript.

  1. Table 2, BMI Z-score, second line, correct 1.030 with 1.036

ANSWER: Thank you very much. We correct the signailed mistake in table 2.
